# How Tall is that Bar Chart?
# Virtual Reality, Distance Compression and Visualizations

Diane Watson[1μα], George Fitzmaurice[2α], and Justin Matejka[3α]

[α]Autodesk Research and [μ]University of Waterloo

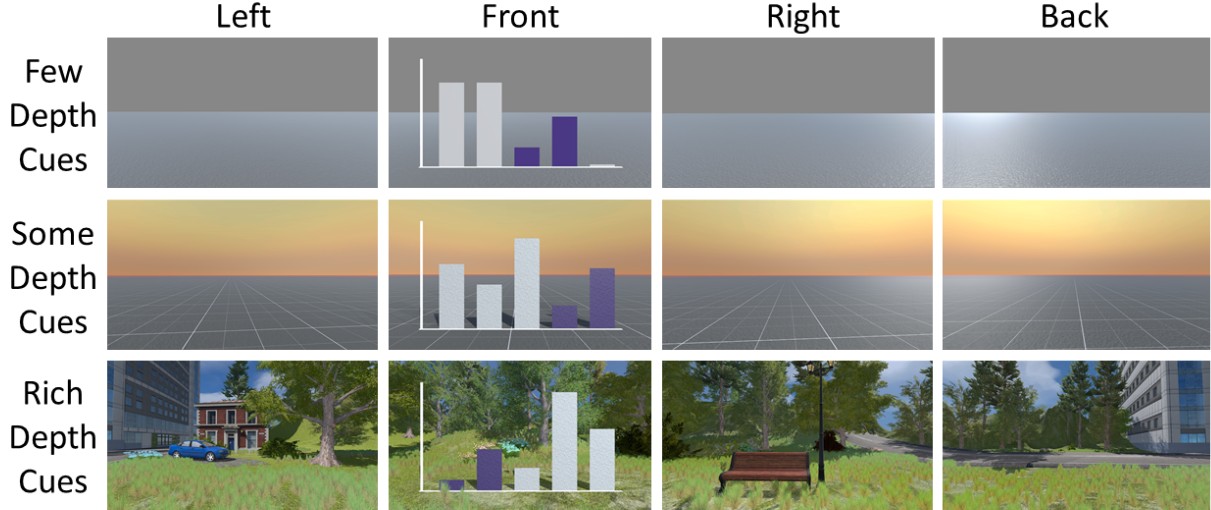

Figure 1. The virtual environments used in Study 1, each with differing levels of depth cues. Participants could look around with the HMD in VR and used the mouse to look around in the screen virtual environment conditions.

## ABSTRACT

As VR technology becomes more available, VR applications will be increasingly used to present information visualizations. While data visualization in VR is an interesting topic, there remain questions about how effective or accurate such visualization can be. One known phenomenon with VR environments is that people tend to unconsciously compress or underestimate distances. However, it is unknown if or how this effect will alter the perception of data visualizations in VR. To this end, we replicate portions of Cleveland and McGill's foundational perceptual visualization studies, in VR. Through a series of three studies we find that distance compression does negatively affect estimations of actual lengths (heights of bars), but does not appear to impact relative comparisons. Additionally, by replicating the position-angle experiments, we find that (as with traditional 2D visualizations) people are better at relative length evaluations than relative angles. Finally, by looking at these open questions, we develop a series of best practices for performing data visualization in a VR environment.

**Keywords**: Visualization; VR.

**Index Terms**: H.5.m. Information interfaces and presentation (e.g., HCI): Miscellaneous

---

[1] dianewatson@uwaterloo.ca
[2] george.fitzmaurice@autodesk.com
[3] justin.matejka@autodesk.com

## 1 INTRODUCTION

As Virtual Reality (VR) technology continues to be developed and expanded, workplace tasks such as viewing information visualizations, are becoming more likely to be executed in a VR environment. While much of the research around traditional screen-based visualizations likely applies to VR, it is unclear how specific VR-related phenomena might alter how effective or accurate these visualizations are. Of particular note, it has been shown that in VR environments people tend to unconsciously underestimate distances, in a phenomenon called distance compression [1, 9, 10, 22, 26, 28, 33, 34, 39, 40, 46]. However, to this point, designers of VR visualizations have not had any guidance about how distance compression will alter visualization effectiveness or user accuracy. For example, even a simple bar chart uses the heights/lengths of the bars to represent data and it is unclear how distance compression will alter one's ability to measure or compare the lengths of the bars.

To solve this problem, we looked to foundational work by Cleveland and McGill which looked at graphical perception of paper-based visualizations [6] and has also been replicated in a digital context [14]. We performed 3 studies, replicating the position-length and the position-angle experiments in a VR environment. Our first study, using the bar chart position-length experiment, provided bar charts in virtual environments both on the screen and in VR and asked participants to measure actual distances (i.e., *the bar is 1m tall*) and relative distances (i.e., *bar A is 80% as tall as bar B*). We explored the suggestion that varying degrees of depth cues could reduce distance compression [10, 12], as well as bar charts of varying scales. In a second study, rather than depth cues we looked at *perspective*, providing participants different ways to move around the look at the various scaled charts in VR. Finally, in the last experiment, we implemented the position-angle experiment, looking at how actual lengths/angles and relative

lengths/angles were measured in bar charts, scatter plots, and pie charts.

Through these studies, we have 5 contributions. First, we confirm the existence of distance compression in VR visualizations but see that is also applies to similarly to screen-based environments. We show that distance compression does negatively affect actual length/distance measurements but may only have a small or negligible effect on relative comparisons. Depth cues had no discernable effect on the accuracy of measurements but do appear to affect one's perception of their *ability* to be accurate. As in traditional 2D visualizations, people are better at relative length evaluations than relative angles. Finally, we provide a set of design guidelines for designers to inform the implementation and creation of effective and useful visualizations in VR.

## 2    RELATED WORK

### 2.1    VR and Visualization

VR visualizations fall under the field of 'Immersive Analytics', though this also refers to augmented reality (AR) and Mixed Reality (MR) visualizations [7]. In the late 90s people were beginning to talk about VR, and visualization, even though systems of the day (mostly VR Caves) did not quite provide adequate capabilities [5, 21, 31, 32, 36]. More recent work provides concrete examples in the environmental [15, 16, 27, 30], medical [11, 19, 48], and archeology [4, 24, 38] domains. A notable example is ImAxes [8] which is a dynamic system were users draw and connect axis in midair in VR allowing for multiple dynamic chart types.

When interacting with a VR visualization, one should keep in mind that multiple views and input modalities may not transfer from the screen to VR [23]. Furthermore, the affordances of an interaction may be different in VR [2]. For example, Simpson et al. found that walking around the dataset in VR was not better than using a controller to rotate it [37]. Cybersickness must also be considered as certain design choices, such as using a controller to move around, may work on a screen but induce nausea in VR [41, 49]. There may be many ways to combat this, for example, Cliquet et al. suggest allowing the user to sit [7].

### 2.2    Visualization and Perception

Cleveland and McGill's foundational work [6] showed that people are better estimating lengths than areas, and better at estimating areas than volumes. Furthermore, people are better at position estimations (i.e., a scatter plot) than angle estimations (i.e., pie chart). These results have been confirmed and replicated more recently by Heer and Bostock with a Mechanical Turk based study [14].

#### 2.2.1    2D vs 3D

The usefulness and effectiveness of 2D visualizations have often been compared to 3D. 2D is considered generally the best approach [35, 44], especially for tasks that require precision [18] and tasks that suffer from perspective distortion (e.g., distance estimation) [17]. However, 3D visualizations can still be useful, particularly when the data has a high levels of detail, structure, and/or complexity (e.g., 3+ dimensions) [17, 18, 25, 45] or when the task involves exploring 3D representations of the real world (e.g., terrain or other real world objects) [17, 18]. 3D may also prove useful by providing ways to explore overlap in network graphs [13, 42, 43].

Now most visualizations can be considered 3D visualizations in VR – even though they might be mapped onto a plane, the ability to look at them from multiple angles might cause occlusion or other problems of perspective. However, we bring up the debate between 2D and 3D graphs because there is some indication that the binocular depth cues provided by modern VR tip the equation in favor of 3D in some situations. When only considering scatterplots, for example, providing binocular cues has shown that 3D visualizations tend win over 2D [25, 29, 45], although this is not always the case [35, 44].

### 2.3    VR and Perception

People tend to underestimate distances in VR using a Head Mounted Display (HMD) [1, 9, 10, 26, 28, 33, 34, 39, 40, 46]. This effect exists even when the VR environment is very similar to, or a recording of the real world [28, 34, 40], and may be partially caused by the limited field of vision of the HMD [9, 22]. Physical factors, such as the weight of the HMD may be also be important [46], especially as the effort perceived to be necessary to walk a particular distance (e.g., if one was wearing a heavy backpack) can have effects perception of that distance [47]. The parameters which affect distance compression have been investigated but are not fully understood. Furthermore, it is unclear how distance compression may affect visualization tasks like comparing two bars in a bar chart.

One possible solution might be a lack of realistic depth cues [10, 12] in VR. Some of these cues, such as light, texture, shape, luminance, linearity of light, object occlusion, motion etc., can be manipulated to be more or less available in a virtual environment. Unlike a 2D screen, VR headsets provide binocular cues because they render a separate image for each eye (although at least one study has suggested that monocular/binocular cues alone are not responsible for distance compression [9]). In fact, current VR headsets, such as the HTC Vive, allow for *most* depth cues that are available in the real world to be implemented in VR [10]. Our first study looks at how fidelity of depth cues might change distance compression when looking at a visualization.

It has also been suggested that this distance compression might be affected by perceptually different distance zones. Armbrüster et al., suggest that distance compression is smaller for objects in peripersonal space (<1m) where an object is within arm's reach [1]. Cutting calls this zone personal space (<1.5m), splitting up larger distances into action space (<30m, interaction of some sort is feasible) and vista space (30m +, further than one would expect to be able to act) [10]. To further investigate these categories of distance, the first two studies use three different sizes of charts, roughly corresponding to personal, action, and vista bar heights.

## 3    STUDY 1 – DEPTH CUES AND SCALE

The chronic underestimation of distances is troublesome for VR, particularly when considering visualizations tend to encode data using absolute or relative distances. Even a simple bar chart uses length to communicate data to the viewer.

Our first study aimed to confirm that visualization in VR are compromised by distance underestimation. Since it has been suggested that more depth cues [10, 12] could lesson underestimation, we designed three virtual environments  with differing levels of *depth-cues*:

### 3.1    Depth Cues

#### 3.1.1    Few Depth Cues

Objects had a consistent luminance and no texture. Shadows were disabled and the sky was a medium gray. A simple textured floor was provided as floating over a void was nauseating in VR. This condition represented a simple chart without embellishments.

#### 3.1.2    Some Depth Cues

Bars now had a slightly crumpled paper texture and responded to the lights in the scene, casting shadows. The floor contained an arbitrary grid and was also textured slightly. Aerial perspective was

applied, allowing distant objects to fade into the sunset sky somewhat. We consider this a 'best practices' chart, with minimal added embellishments, all of which directly contribute depth cues to the environment.

### 3.1.3 Rich Depth Cues

In addition to texture, luminance, and aerial perspective, the scene was augmented with objects that could be used to determine relative sizes. Trees, a light post, some bushes provided general cues about scale. A house, car, and park bench were also in the scene, as these have relatively standard sizes. Similarity, a skyscraper was in the scene, as a floor of a building is also about the same size. These objects were not immediately in view, the participant would have to look at them directly. Grass and flowers on the ground provided cues of relative density. This condition, while being very rich with depth cues, was also a bit extreme; one can imagine that not every visualization has a place for trees, cars, and buildings (Figure 1). However, Bateman et al [3] showed that embellishments that add context to the visualization can improve memorability, and given the prevalence of infographics, it is not impossible that some visualizations might provide relevant, contextual objects (e.g., a visualization about deforestation could contain trees).

## 3.2 Scales

We were also interested in measuring this effect at multiple *scales*. The corresponding chart heights and task specific bar lengths can be seen in Table 1. In all conditions, the participant viewed the visualization from 4m back:

### 3.2.1 Personal scale

The entire visualization could be seen at one time when looking straight ahead without looking significantly up or down.

### 3.2.2 House Scale

The larger visualization required that the participant need to look up somewhat to see the entire chart.

### 3.2.3 Skyscraper Scale

The visualization was extremely tall, requiring the viewer to tilt their head back and look way up. While a barchart as high as a skyscraper is *unlikely* to be very useful, we included this scale because for very large or complex visualizations it is possible that a user could to navigate to a view where some of the data is very far away.

Finally, we compared VR with an on-screen condition which featured virtual environments with the same scales and depth-cues (without, of course, the binocular depth cues provided be the HMD). This gave us a scale (3) by depth-cues (3) by screen/VR (2) factorial study. We also added a real-world condition, with a simple bar chart contained on a monitor. This was to provide a baseline as it was similar to the foundational work done by Cleveland and McGill [6].

## 3.3 Task

Cleveland and McGill [6] provide several tasks for evaluating perception of lengths in a visualization. We chose to mimic their position length experiment, specifically using their Type-1 task (as this had the lowest error). This task provides a 5 value bar chart, with two side by side bars marked with a dot which have percentage differences ranging from 18% to 83%. The participant is asked to evaluate, without explicitly measuring, what percentage the smaller bar is of the larger. Our task mimics theirs, down to the way they chose relevant values for the bars in the task, except that we used a single bar chart instead of two side by side bar charts. We also colored the bars of interest because for dot at the bottom would be

insufficient for differentiation when looking way up in the *skyscraper scale*.

Participants completed 7 blocks (*depth-cues* (3) x s*creen/VR* (2) + 1 *real-world*), counterbalanced with a Latin square design. In each block they viewed 18 bar charts (126 total), 6 of each *scale*, in random order, except in the real-world condition where all 18 bar charts fit on the screen at 30 cm tall. Using [6]'s template as a guide, we randomly generated sets of 6 tasks; each set containing two tasks with a percentage difference between 10% and 40%, two between 40% and 60% and two between 60% and 90%.

For every bar chart, participants were asked to specify the percentage the smaller bar was of the larger, and the absolute height of the smaller bar in the virtual environment (or the real-world height on the screen, in the *real-world* block). While relative comparisons (bar compared to axis) is indeed the more common visualization task, we also asked about actual heights as this was relevant to the distance compression literature and is a relevant visualization task in situations (e.g., terrain map where 1m = 1km in the real world). Participants were told not to walk around in VR, but could look around in any direction. In the on-screen virtual environment, participant could not move, but could look around using the mouse. Their 'virtual head' was placed at the same height as their real head had been in VR.

Like in the original position-length experiment [6], participants were instructed to not explicitly measure (e.g., using a finger) or explicitly calculate distance. Also, because all participants in the pilot study expressed that the task was too hard and that they had no confidence in their estimations, we included an instruction that the task was supposed to be hard and not to feel discouraged.

## 3.4 Measures

Before the study participants filled out a demographics questionnaire asking about VR and game experience. Participants responded verbally when asked for percentages and heights. Answers were recorded and later merged with the logged study data. After the study, they filled out a simulator sickness questionnaire (SSQ)[20] and a questionnaire asking them whether they thought they performed better when estimating percentages or heights, which *depth-cues* virtual environment they thought they performed best in, whether they were better *screen/VR*, and finally were asked to rate each of the 7 blocks in terms of how well they thought they performed.

Like in the original position-length experiment [6], we took the log error of the percentage estimations

$$LogErrorP = log_2\left(\left|percent_{guessed} - percent_{actual}\right| + \frac{1}{8}\right)$$

For the height estimations, we calculated the error as a percentage of the actual height they were estimating. This meant if the bar was 20 m tall, and the participant guessed either 18 m, or 22m, they had a height error of 10%.

$$ErrorH = \frac{\left|height_{guessed} - height_{actual}\right|}{height_{actual}}$$

## 3.5 Participants

Study 1 had 18 participants recruited, with one removed for not following the instructions consistently, and one removed as an outlier with results more than 2 standard deviations from the mean, resulting in data for 16 participants. Details about the participants can be found in Table 1. Participants were remunerated with a 25 CAD gift card.

## 3.6 Results

Results were analyzed with two (*depth-cues* (3) x *screen/VR* (2) x *scale* (3)) RM-ANOVAs (the *real-world* condition was not part of this RM-ANOVA, instead providing a sanity check that our participants performed similarly to [6]. Overall measurement means and standard deviations can be found in Table 1 and charts are in Figure 2.

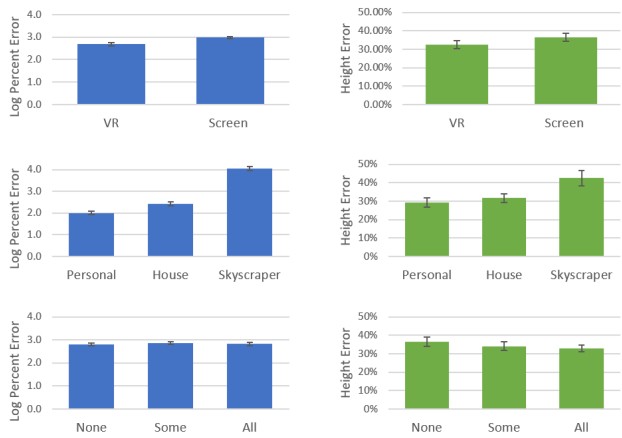

Figure 2. Log percent error (left) and height estimation error (right) for screen/VR (top row), scale (middle row) and level of depth cues (bottom row) used in Study 1. *(Note: Error bars show standard error.)*

### 3.6.1 Percentage Log Error

There was a main effect of s*creen/VR* (F(1,15) = 27.63, p < 0.01). When given the same virtual environments on the screen and in VR, participants had less error in VR (Figure 2). There was no main effect of *depth-cues* (p=.53). There was a main effect of *scale* (F(2,30) = 185, p < 0.01) (Figure 2). People were significantly worse at larger distances.

There was an interaction effect of *screen/VR* x *scale* (F(2,30) = 32.49, p < 0.01). At the *personal* scale, screen-based virtual environments has less error than VR, but this reversed at the larger scales.

The *real-world* condition had a mean log percent error of 1.7 (SD: 0.94), which is slightly higher, but very similar to value achieved by the original paper based task [6] which was 1.5.

### 3.6.2 Height Error

As expected, 77.5% of all height evaluations were underestimations. There was a main effect of *screen/VR* (F(1,15) = 5.80, p < 0.05)*,* with participants having less error in VR. There was a main effect of *scale* (F(2,30) = 5.403, p < 0.05), with higher error at larger scales. There was no main effect of *depth-cues* (p=.15) and no interaction effects.

The *real-world* condition had a mean height error of 20% (SD: 16%).

### 3.6.3 Subjective Rankings

Participants indicated whether they were better in VR (62%), the screen-based virtual environment (19%) or performed equally well on both (19%). Most participants indicated that they performed best in the *rich* virtual environment (81%), while a few indicated they were best in the *some* virtual environment (19%). The 7 blocks (*depth cues* (3) x s*creen/VR* (2) + 1 *real-world*), sorted by mean participant rank, are: *rich/VR* (1.6), *some/VR* (3.0), *rich/screen* (3.1), *some/screen* (4.5), *real-world* (4.7), *few/VR* (4.9), *few/screen* (6.1). Although we did not formally record participants with audio or video, we tried to take notes if they commented on the

helpfulness (or lack of helpfulness) of a virtual environment during or after the study. Most participants commented that the task was very hard, particularly with heights (e.g., *"I don't think I am very good at this"*, *"I have no idea how tall that is"*). Every single participant commented at least once about some facet of the *rich* depth cue conditions as helpful (e.g., *"The trees help"*, *"I like the building, I can count the floors"*, *"How big is a house, that chart is as big as a house"*).

### 3.7 Summary of Results

In general, people were quite good at evaluating percentages, but poor at evaluating heights. They were better in VR, perhaps due to binocular depth cues, or due to physical sensations such as tilting one's head back to look up. Scale was as expected, important, with larger distances resulting in more error in both measurements. Depth cues did not seem to be influential in either height or percentage evaluations. This first study confirmed that distance underestimation occurs in a visualization context, in this case bar charts, in VR. However, the results suggest that percentage estimations are not nearly as negatively affected as height estimations. Participants were off by 9.7% on average, which is very small when considering that most answers were given as a multiple of five, introducing an expected error of 2.5%. Furthermore, VR had less error than the equivalent screen-based virtual environments, meaning that VR might be a better option than a screen-based visualization where one needs to look around, at least in some situations.

Subjectively, people felt they were more accurate at percentages and better in VR. However, even though we did not find a significant difference between the different *depth-cue* conditions, people collectively *felt* that they were more accurate in the rich depth cue conditions. This is interesting because it means that while depth cues might be less impactful on task performance, it does seem to be impactful on user comfort and their own perception of competency.

## 4 STUDY 2 – MULTIPLE PERSPECTIVES

In this study we were interested in how perspective and motion would change perception of distances. In particular, the fixed position near the base of the bar chart used in Study 1 meant that for the larger scales of charts, users would experience significant perspective issues such as foreshortening.

Other than providing movement and the ability to take a new perspective, we also made a few changes to Study 1. Since we found no effect of *depth-cues* on task performance, we fixed this factor at our sunset-like, *some* depth cues virtual environment which we consider a reasonable best practice. Despite the preference of participants for the park-like *rich* depth cue environment, we acknowledge that complex context-rich settings full of trees and buildings may not be universally suitable for all visualizations. The scales we used in this study did not change, however since we were interested in letting participants take perspectives that were possibly far away, we made our bar charts 1.5x as wide to be more visible from a distance. This meant that we moved the front and center starting position back 1.5 m such that at the *personal* scale the participant would still see the whole chart. We then calculated, using this front location, two other fixed positions (15m to the left & right), and a variable back position such that the view was elevated and far enough away that both colored bars could be seen in their entirety. This back position was different in every bar chart and provided the participant with perspective that did not require them to look up or down and removed foreshortening effects.

Unlike the first study, where participants stood on the ground, here participants stood on virtual platforms (Figure 3). These hexagonal platforms featured transparent glass railings, interaction

Table 1: Study Details

| | | Study 1 | Study 2 | Study 3 | |
|---|---|---|---|---|---|
| **Task Type** | | Position-Length [4] | Position-Length[4] | Position-Angle[4] | |
| **Scale** | | | | *Bar/Scatter* | *Pie* |
| *Personal* | *Height* | 3m | 3m | 13m | 5m |
| | *Task Bar Height* | < 1.7m | < 1.7m | < 5m | - |
| *House* | *Height* | 30m | 30m | 30m | 12m |
| | *Task Bar Height* | < 17m | < 17m | < 12m | - |
| *Skyscraper* | *Height* | 180m | 180m | - | - |
| | *Task Bar Height* | < 100m | < 100m | - | - |
| **Participants** | *Age* | M 35.0, SD: 10.7 | M: 34.5, SD: 11.0 | M: 28.5, SD: 4.3 | |
| | *N* | 16 (4 female) | 10 (2 female) | 10 (3 female) | |
| | *Measurement* | 4 used metric | 5 used metric | 6 used metric | |
| | *VR Experience* | 11 tried before, 3 very familiar, 2 experts | 3 no experience, 2 tried before, 5 very familiar | 7 tried before, 2 very familiar, 1 expert | |
| **Measures** | *Log Percent Error* | M: 2.67, SD: 1.37 | M: 1.96, SD: 1.18 | M: 1.85, SD: 1.67 | |
| | *Height Error* | M: 32%, SD: 23% | M: 34%, SD: 22% | M: 34%, SD: 27% | |
| | *Angle Error* | - | - | M: 20%, SD: 16% | |
| | *Simulator Sickness* | M: 5.1, SD: 4.5 | M: 10.1, SD: 6.2 | M: 6.2, SD: 4.8 | |
| | *Performed Better At* | percents (88%), heights (0%), both (12%) | percents (90%), heights (0%), both (10%) | percents (50%), heights (10%), both (40%) | |

instructions, and lights that would light up when the participant stood on a centrally located pressure plate. (Participants were asked to return to the center between tasks). The width of the platform (about 2m) corresponded to the maximum walkable space as calibrated by the HTC Vive. This meant that participants could always walk around on a platform and even lean over the railings safely. Other than being functional in terms of movement, the consistently sized platforms could be used for relative sizing, like the objects in the *rich* depth cues virtual environment.

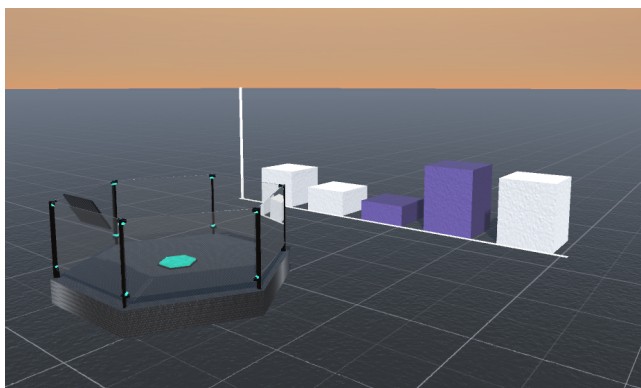

Figure 3. Front platform near personal scale bar chart. Participants would view the world from the center of the platform. In some conditions platforms functioned as elevators.

Participants were always on a platform, however, we provided the following movement modes.

**Front Platform Only:** Like our naïve perspective chosen in the first study, the participant was stuck at the front and bottom of the chart. The participant could not teleport to a new location or move the platform up or down. However, unlike the first study, participants could walk around on the platform.

**Back Platform Only:** The participant started on a variable location back platform and could walk around but not teleport or use the platform elevator.

**Teleport Anywhere:** Participants started at the front location and could teleport/move the platform they were on by pressing the trigger, aiming a visible arc pointer to a valid location (marked by a repeating blue pattern) and then releasing the trigger. Participants could teleport anywhere in a 100m square centered on the chart. Participants could not move so close to the chart that they intersected it (invalid area was marked in a red). The elevator

platform could be moved up and down by using a diegetic interface on the touchpad.

**Teleport 4 Platforms**: Participants could teleport to front, left, and right, fixed location, elevator platforms as well as the variable location back platform. Additional platforms are only seen when teleporting so they do not block the chart. Platforms were selected by aiming the arc pointer directly at, near, or in the general direction of a platform.

**Teleport Front/Back**: Participants could teleport like in the Four Platform condition, but could only access the front and back elevator platforms.

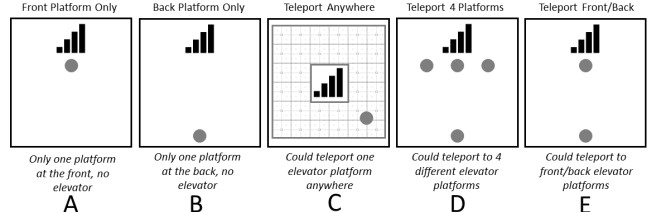

Figure 4. Types of movement allowed in Study 2.

### 4.1 Task

Tasks were generated the same as they were in Study 1. Participants completed 5 counterbalanced blocks, one for each movement type. Each block was introduced in a training mode where participants could try out the relevant movement interactions. During the study tasks, participants were asked to teleport at least once before giving their answers (if applicable). They were asked to return to the center of the platform in between each of the 90 tasks.

### 4.2 Measures

The measures employed were similar to Study 1, except that the final questionnaire asked them to rank their performance with the 5 movement types.

### 4.3 Participants

Study 2 had 11 participants, one was excluded as they were a clear outlier (more than two standard deviations from the mean). All participant details can be found in Table 1. The study took one hour and participants were remunerated with a 25 CAD gift card.

### 4.4 Results

We performed two (*movement-type* (5) x *scale* (3)) RM-ANOVAs. Overall measurement means and standard deviations are in Table 1, and charts are in Figure 5.

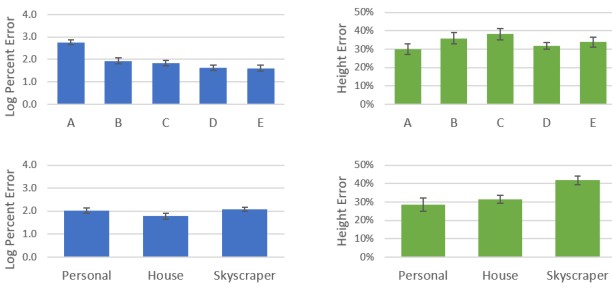

Figure 5. Log percent error (left) and height estimation error (right) for each movement type (top row) and scale (bottom row) used in Study 2. *(Note: error bars show standard error.)*

### 4.4.1 Percentage Log Error

There was a main effect of *movement-type* ($F_{(4,36)}$ = 51.6, p < 0.001). Post-hoc tests showed that Front Platform Only was significantly worse than all other conditions. There was no effect of *scale* (p=0.12) and no interaction effects. Overall measure averages can be found in Table 1.

### 4.4.2 Height Error

There was no main effect of *movement-type* (p = .31) but there was an effect of *scale* ($F_{(2,18)}$=6.7, p<0.01). Post hoc tests showed that the largest scale lead to significantly higher error than the smallest and medium scale (p<0.05).

### 4.4.3 Subjective Rankings

The movement-types, sorted by mean rank, are: continuous-teleport (1.8), teleport-front/back (2.0), teleport-4-platforms (2.1), back-platform-only (4.1), front-platform-only (4.7).

## 4.5 Summary of Results

Adding movement/different perspectives to the viewpoint used in the first study always resulted in improvements to percentage estimations. The log percentage error of these improved conditions is about 1.7 which is very close to the 1.5 log percentage error achieved by Cleveland and McGill's position-length type 1 task which we modelled our study after. Furthermore, the effect of *scale* on percentage estimations was essentially eliminated when participants had the opportunity to view the visualization from far away and view the entire set of bars at once. Thus, it appears that relative distances tasks, like the percentage estimation we used here, are robust to the perceptual effects of VR if one can view the chart from different perspectives.

On the other hand, directly estimating the height was still problematic when movement and perspective were added. There was no condition which improved people's height estimations and larger scales were still more difficult. Thus movement/perspective was not successful at improving people's ability to estimate heights.

## 5 STUDY 3 – OTHER CHART TYPES

Now that we have established, at least when it comes to bar charts, estimating relative distances may be robust to the effects of distance compression in VR, we were interested in looking at this effect in other chart types. To this end we ran a third study using bar charts, pie charts and scatter plots (Figure 6). We used the sunset environment from study 1 and the teleport anywhere movement from study 2 as it was ranked the highest. Also, because the first two studies confirmed that people are bad at estimating the size of skyscrapers, we had 9 of the 18 charts in each condition fit all data directly ahead without needing to look up and the other 9 about twice as tall.

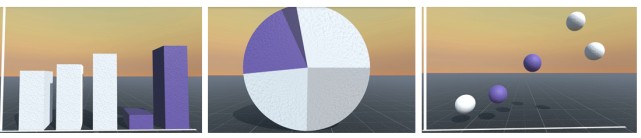

Figure 6. Chart types used in Study 3.

**Bar Chart**: Like with the bar chart in the previous studies, participants were asked to make percentage estimations between two colored bars and to estimate the height of the smaller colored bar. Colored bars were always consecutive but their order was randomized.

**Scatter plot**: Instead of bars, this chart used spherical markers. The markers were spread out horizontally on the x-axis such that they were all between 0.5m and 2.5 m apart, however, the colored markers were always consecutive and 1.5m apart. Participants were asked to make percentage estimations between the colored markers with respect to height (y-axis) and to estimate the height of the shorter colored marker.

**Pie Chart**: This five-section pie chart had two colored segments and three distinctly colored white sections. Colored segments were always in a random consecutive position and were assigned either dark or light purple randomly. Like the Cleveland and McGill's [6] position-angle experiment, participants were asked to make percentage estimations between the colored segments. However, as a pie chart uses angles rather than heights to encode data, participants were instructed to estimate the angle of the smaller segment.

## 5.1 Task

Tasks were generated similar to Cleveland and McGill's [6] position-angle experiment. This meant that 5 numbers summing to 100 were generated, with percentage differences ranging from 10% to 97%. Participants completed 3 counterbalanced blocks, one for each chart type. Each block was introduced in a training mode where the researcher walked them through the exact questions used in this task. During the study tasks, participants were asked to teleport, walk around, or use the elevator at least once before giving their answers. They were asked to return to the center of the platform in between each of the 54 tasks.

## 5.2 Measures

The measures employed were similar to Study 1 and 2, except that the final questionnaire asked them to rank their performance with each chart.

Additionally, to compare participant's angle estimations to height estimations, we used a very similar formula to calculate the angle estimation error as a percentage of the actual angle they were estimating.

$$ErrorA = \frac{\left|angle_{guessed} - angle_{actual}\right|}{angle_{actual}}$$

## 5.3 Participants

Study 3 had 10 participants. All participant details can be found in Table 1. The study took 40 minutes and participants were remunerated with a 25 CAD gift card.

## 5.4 Results

We performed a (*chart-type* (5) x *scale* (2)) RM-ANOVA. Overall measurement means and standard deviations can be found in Table 1 and charts are in Figure 7.

### 5.4.1 Percentage Log Error

There was a main effect of *chart-type* ($F_{(2,18)}$ = 11.06, p< 0.001). Post-hoc tests showed that Pie Charts were significantly worse than

all other condition (p<0.05). There was no effect of *scale* (p=0.30) and no interaction effects.

### 5.4.2 Height and Angle Error

There was a main effect of *chart-type* (F(2,18) = 11.39, p < 0.01). Post hoc tests showed that angle estimations had significantly less error than heights (p<0.01). There was no effect of *scale* (p=0.65) and no interaction effects.

### 5.4.3 Subjective Rankings

The *chart-types*, sorted by mean rank, are: *bar* (1.4), *pie* (2)*, scatter* (2.7)*.

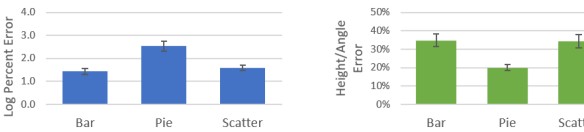

Figure 7. Log percent error (left) and height estimation error (right) for each chart type in Study 3. *(Note: error bars show standard error.)*

## 5.5 Summary of Results

When it comes to percentage estimations, our results mirror Cleveland and McGills [6] results: people are better at lengths than they are at angles, by a factor of 2.1 (1.96 in the original study). This result suggests that for percentage estimations, other perceptual tasks (e.g., area) should follow the same patterns in VR as the original work.

However, when looking at angle/height estimations, participants were better at angles. This could be because one only needs to look at the innermost pie chart point to do this estimation (as opposed to looking at the bottom and top of a large bar chart), because all angles are bounded by 360 (max angle was 100 degrees in our study), or because the angles were relatively small. Future work should investigate this more closely, especially at larger and smaller scales.

## 6 DISCUSSION

This work suggests that the distance compression problem that occurs in VR does alter one's perception of data visualizations. However, relative distance tasks, like the percentage estimation tasks in these three studies, appear to be robust to this distance compression, particularly when participants can reach a perspective where they can view the chart from far back.

## 6.1 Design Guidelines

### 6.1.1 VR is Good for Virtual Environments

In Study 1, VR had less error than the equivalent screen-based virtual environment for both percentage and height estimations. While VR may not be better in all circumstances (e.g., flat screen image), when the visualization exists inside a virtual environment, VR can be a good choice for immersive analytics.

### 6.1.2 Use Movement Modes that Avoid Nausea

Unity's practitioner guidelines [41] recommends that one avoids user simulator sickness by designing to avoid vection. Vection occurs in VR when the user's vestibular system is receiving different signals than their eyes and ears. This occurs most often in VR when the movement modality causes the player to experience motion in VR that their body does not (e.g., mapping movement in VR to a thumbstick on a controller), or when experienced bodily motion has no effect in VR (e.g., not updating the VR environment when the user turns their head).

Although we were not specifically investigating or avoiding nausea, we did find some techniques we used successful. We used a fade in/out teleport mechanism and platforms that matched the safely walkable space in the real world in Study 2 and Study 3 to avoid vection. The elevator feature was unfortunately vection inducing because it was activated with the touchpad instead of equivalent bodily motion. We combated vection-related nausea by severely limiting the elevator speed and used an easing function to prevent sudden stops. A future implementation of an elevator might have 'floors' that can be accessed with a teleport-like fade effect.

### 6.1.3 Encode Data with Relative Distances

One should not have any requirements or expectations that a user can estimate a distance in VR. In all three studies, participants were underestimating heights on average by 33%, (i.e., 2/3 their actual value). However, across all three studies, participants provided extremely low-ball heights 15% of the time, underestimating by more than 50%. Conversely participants were much more accurate in the percentage estimation task, off only by 6-10% on average across all three studies.

Therefore, designers should encode data in ways that allows users to compare two distances/lengths rather than expecting them to measure a distance directly. In a simple graph like a bar chart or scatter plot, this can be as simple as providing a labelled axis immediately beside the data. Avoid, say, providing a scale for a map requiring that one can estimate a distance (e.g., 1 inch = 1 mile). We also recommend that where a measurement of distance is necessary, one should provide a tool which can be used for comparison, like a ruler, or a measuring tape.

### 6.1.4 Consider Maximum Scale

It is important to consider the most extreme perspective that the user can navigate into, or, rather to consider the maximum distance from themselves that a user might be asked to evaluate. In Study 1 participants were pretty good at the personal and house scale, and were predictably bad at the skyscraper scale. Therefore, we recommend that one create situations that expect users to be evaluating distances less than 15m, though this is just a rough estimation based on the particular distances we used in the study. Future work is needed to provide a better guideline.

### 6.1.5 Provide an Overview Perspective

In Study 1, larger scales meant higher error in both percentage and height estimations. However, when the ability to view the data from an overview perspective was added in Study 2, this effect disappeared for percentage estimations.

If one does use large distances, or data that is far away from the user, providing a way for the user to make a quick overview of relevant data can remove the negative effects of large distances by removing problems like foreshortening. It should be noted that in Study 2, the *teleport-anywhere* condition where one could move their view to an overview perspective was not significantly better than the *back-platform-only* condition which automatically provided an overview perspective. Therefore it may be enough to simply provide a generated overview of your visualization, or one could provide freeform movement options like *teleport-anywhere* depending on your needs.

### 6.1.6 Users Appreciate Additional Depth Cues

Given that every participant mentioned how helpful additional depth cues were in Study 1, despite no change in task performance, we would recommend providing as many depth cues as possible to improve user's comfort and perceived competency with the task. This could mean simple things like adding texture to an object, or more complicated, fully developed, contextually relevant environments with relatively sized objects.

## 7 CONCLUSION

In this paper we investigated how the phenomenon of distance compression alters perception of visualizations in VR. Through three studies that replicate foundational work around the perception of visualizations we found that estimations of actual lengths, in this case the heights of bars in a bar chart, are negatively impacted by distance compression, but relative distances are not. Furthermore, as with traditional visualizations, people can better estimate relative lengths over relative angles, suggesting that much of the existing perceptual research on visualizations may still apply. Finally, we provide set of design guidelines for designers wishing to develop VR visualizations that limit the negative effects of distance compression.

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
