# OpenReview forum: "How Tall is that Bar Chart? Virtual Reality, Distance Compression and Visualizations"
_graphicsinterface.org/Graphics_Interface/2021/Conference/Second_Cycle — GI 2021_

### Official Review · Reviewer_pcMX · 2021-05-01
**Well-executed experiments, could have been more exploratory**

**Rating:** 6
**Confidence:** 2

**Review:**

This work studies the effect of VR perception distortions on the accuracy of data visualization tools, such as bar charts, in a VR environment.

The strength of this paper is the well-executed studies with good statistical analysis as well as a set of clear recommendations for practitioners who want to design VR-based data visualizations.

The paper, however, could have been more exploratory. The main question of the paper is, in my opinion, a simple one. I am far from an expert in visualization, but I know that relativity is key to our perception. So if we preserve relativity in VR, we will most likely benefit from it for accurate perception in this environment as well.

I think the paper is clearly written. I spotted a few typos here and there:
- In general, I think the reference bracket should come immediately after "et al.".
- Page 2, Column 1, Paragraph 2: "As-in" -> "As in"
- Page 2, Column 1, Paragraph 3: "Visualizations" -> "visualizations"
- Page 3, Column 1, "Few Depths Cues" paragraph: "embellishments.." -> "embellishments."
- Page 3, Column 2, Paragraph 1: "visualization s" -> "visualizations"

---

### Official Review · Reviewer_gpJy · 2021-05-04
**Confirming important phenomena in VR**

**Rating:** 8
**Confidence:** 4

**Review:**

This paper presents a series of studies replicating the stimulus judgement tasks of Cleveland and McGill in a variety of VR scenarios. The three experiments respectively investigate the effects of depth cues and scale, the viewing position and viewing freedom, and the measurement of phenomena in other types of charts. The results showed that measuring absolute scale remains problematic, especially at larger scales, that the depth cues did not have an effect, but participants preferred more cues, and viewing freedom improved results. The effects were consistent in a brief experiment on other visualization types.

Overall, I found the studies to be solidly executed and reported. The paper was clearly written with only a few typos. The literature review was thorough and covered both related experiments in the visualization literature and issues of designing effective and comfortable VR environments. The findings are confirmatory in nature, but this is valuable. The discussion expands on the findings with useful takeaways for the design of immersive visualizations.

The reporting of study 3 was a bit brief. It was not clear to me what level of movement constraints were applied. Was the movement constrained at all, or unconstrained similar to condition "C" in Study 2? If so, why use this method, instead of D or E?

Typos:
- abstract: will be increasing used -> increasingly
- introduction: paper based -> paper-based
- p5: to me more visible -> to be more visible
- p8: because the on only needs -> because one only needs (?)

---

### Official Review · Reviewer_HFYH · 2021-05-04
**Potentially nteresting work but lacking rigour in study design and analysis**

**Rating:** 5
**Confidence:** 5

**Review:**

This paper describes a series of 3 user studies aimed at exploring the effects of scale on data visualisations in VR.

This work addresses a current topic of how new immersive displays impact our abilities to perceive data visualisations. In particular, the studies focus on the known issue of distance underestimation, and explores how this applies to potential vis applications.

Overall the work is clearly explained, although some important details are missing. Similarly, the studies produce some potentially useful results, but the design and analysis are lacking rigour in some aspects. For instance, the few/some/rich depth cues conditions in study 1 make sense in principal, however, there are no specific details about exactly which of the many known depth cues are targeted here - it's not clear that specific depth cues were considered in this design.  Similarly, the logic behind the design of the available viewing locations in study 2 seems a bit arbitrary. I'm not sure that such a large area of 100 m square is necessary to achieve the required effect, and there is no clear reason behind the chosen locations. The far location is described as "far away", but there is not enough detail provided for repeatability.

In the analysis, it is problematic to take the absolute value of the estimations. Since the literature shows a tendency to underestimate, there is no reason to assume over-estimations and under-estimations are equivalent. I believe the results should be analysed with a two-tailed test that differentiates between these, rather than a single-tailed test.

The removal of participants based on 2SD is also a bit aggressive. It may be that the removed participants appear as outliers due to the relatively small sample size of the study. There are also no post-hoc test results provided for the ANOVAs, except for some reason in study 3.

While the first 2 studies provide some interesting results, I don't think the third study adds much. There are no variables related to scale or viewing condition here, so the study is really about comparing different representations. We don't learn anything new here that we didn't already know from similar previous studies in 2D.

In general the paper is on the right track but the contribution could be improved with further attention to detail in the design and analysis. For this reason I'm leaning a bit on the side of rejection, since another cycle seems to be needed here.

minor issues:
- there is some inconsistency in terms - e.g. condition C in study 2 is sometimes called teleport anywhere and other times continuous teleport. Adding the lettered labels A,B,C,... further adds to this inconsistency.
- It would be helpful to add a label to the x-axis of the bar charts, rather than require the reader to look for these in the caption, or to rely on the condition labels alone.

---

### Meta-Review · Area_Chair_cgDW · 2021-05-06

**Recommendation:** Accept
**Confidence:** 4

**Metareview:**

The reviewers were somewhat split regarding this manuscript, finding the paper alternatively to be relevant to a current topic and clearly explained, but alternatively narrow in scope and having some problems with the specification of experimental conditions and the analysis of the results.

Given the scores, the paper is marginally acceptable. The authors should carefully review the issues raised by the reviewers, in particular, better justify the exclusion of participants over 2SD and the suggestion that a two-tailed test may be more appropriate given that under and over-estimation cannot be assumed to be equally likely. The specific selection of the conditions for the second study need to be justified better. Reviewers agreed that the third study, while raising an interesting question, was too brief and superficial to be useful for any conclusions.

That said, the topic is relevant and the question is of interest as immersive analytics is a growing field of interest. The discussion gives takeaways that could be useful for VR pratictioners.

---

### Decision · Program_Chairs · 2021-05-08

Accept